# Suppressing Pro-Apoptotic Proteins by siRNA in Corneal Endothelial Cells Protects against Cell Death

**DOI:** 10.3390/biomedicines12071439

**Published:** 2024-06-27

**Authors:** Susanne Staehlke, Siddharth Mahajan, Daniel Thieme, Peter Trosan, Thomas A. Fuchsluger

**Affiliations:** 1Department of Ophthalmology, Rostock University Medical Center, 18057 Rostock, Germany; peter.trosan@med.uni-rostock.de; 2Institute for Cell Biology, Rostock University Medical Center, 18057 Rostock, Germany; 3Department of Ophthalmology, University of Erlangen-Nuremberg, 91054 Erlangen, Germany; 4Institute of Polymer Materials, University of Erlangen-Nuremberg, 91058 Erlangen, Germany

**Keywords:** corneal endothelial cells, donor cornea, siRNA, anti-apoptotic Bax and Bak, apoptosis, caspase-3, TUNEL, Annexin V, confocal laser scanning microscopy

## Abstract

Corneal endothelial cells (CE) are critical for the cornea’s transparency. For severe corneal damage, corneal tissue transplantation is the most promising option for restoring vision. However, CE apoptotic cell death occurs during the storage of donor corneas for transplantation. This study used small interfering (si)RNA-mediated silencing of pro-apoptotic proteins as a novel strategy to protect CE against apoptosis. Therefore, the pro-apoptotic proteins Bax and Bak were silenced in the human corneal endothelial cell line (HCEC-12) by transfection with Accell™siRNA without any adverse effects on cell viability. When apoptosis was induced, e.g., etoposide, the caspase-3 activity and Annexin V-FITC/PI assay indicated a significantly reduced apoptosis rate in Bax+Bak-siRNA transfected HCECs compared to control (*w*/*o* siRNA). TUNEL assay in HCECs exposed also significantly lower cell death in Bax+Bak-siRNA (7.5%) compared to control (*w*/*o* siRNA: 32.8%). In ex vivo donor corneas, a significant reduction of TUNEL-positive CEs in Bax+Bak-siRNA corneas (8.1%) was detectable compared to control-treated corneas (*w*/*o* siRNA: 27.9%). In this study, we demonstrated that suppressing pro-apoptotic siRNA leads to inhibiting CE apoptosis. Gene therapy with siRNA may open a new translational approach for corneal tissue treatment in the eye bank before transplantation, leading to graft protection and prolonged graft survival.

## 1. Introduction

The corneal endothelium (CE), a monolayer of hexagonal cells located on the posterior corneal surface, regulates the water balance for optimal transparency and visual acuity [1,2]. CEs represent a cell or tissue type that challenges the researcher due to their cellular profile, as CEs are characterized by cell cycle arrest [3,4]. Due to their barely dividing character, all corneal endothelial cell diseases cannot be cured by biological regeneration. Genetic and environmental factors, e.g., can cause endothelial dysfunction [5,6,7]. Corneal diseases are among the leading and most common causes of visual impairment [8,9,10]. With a pronounced reduction in CE density, the physiological protection of the corneal stroma against swelling due to the aqueous humor can no longer be guaranteed, and corneal opacity occurs [11]. The clinical consequence is a functional loss of visual acuity, which can lead to blindness [12].

In many cases of severe corneal damage, corneal tissue transplantation (keratoplasty) is the only promising option for rapid visual rehabilitation [13,14]. The number of corneal transplants worldwide is over 180,000 per year (2018), including over 8000 in Germany (2021) [15,16,17]. Unfortunately, given the global demand for corneas for transplantation, access is limited in many parts of the world, the number of donor corneas is low [18], and the rejection rate is very high at an average of 32 ± 20% per eye bank (2021) [16]. The risk of graft failure increases with time after transplantation, so long-term visual acuity continuously decreases after transplantation or remains very low, necessitating re-transplantation [19]. A significant factor in this discrepancy is the loss of CE caused during storage of donor corneas in eye banks and by mechanical stress (during enucleation or keratoplasty) (Figure 1) [16,20]. The most important reason for rejection is the inadequate endothelial quality (48%) [16]. Significant loss of CE through apoptosis renders the tissue untransplantable, as the density of corneal tissue falls below the minimum threshold of 2000 cells per mm^2^ [16].

The cultivation of donor corneas is of interest as they serve as transplants. Due to the storage of the corneas in the eye bank for up to 4 weeks [20], under established protocols in cell culture flasks in an incubator, the CE is easily accessible for gene and cell therapies, and the effect of the gene transfer can be well visualized due to the transparency of the ex vivo cornea [16]. Prevention of CE apoptosis, as a critical factor, could be an essential tool to improve graft quality and increase the amount of donor tissue available for transplantation (Figure 1) [5,17,21,22,23,24].

Apoptotic processes have been studied in detail as a cause of damage to the CE [25,26,27,28]. Apoptosis, programmed cell death, maintains tissue homeostasis [29]. Controlled apoptosis, regulated by signaling pathways, activates a caspase signaling cascade that destroys proteins, cell structures, and DNA [30,31]. One of our studies focused on gaining a detailed insight into upstream and downstream apoptotic proteins that serve as apoptosis markers in both corneal cells and corneal tissue [32].

Whether a cell will survive or die due to apoptosis is determined by the balance of activity between pro- and anti-apoptotic members of the B-cell lymphoma 2 (Bcl-2) superfamily [30,33,34]. Strategies to prevent apoptosis depend on which apoptosis pathway proteins are targeted [17,24,27,35,36,37]. One possible approach to disrupting the protein balance in a cell is the selective knockdown of pro-apoptotic proteins, such as Bax (BCL2 associated X, apoptosis regulator) and Bak (BCL2 antagonist/killer 1), in order to prevent the activation of the apoptosis cascade and thus finally apoptosis [27,31]. Silencing with small-interfering RNAs (siRNAs) is a powerful, particular tool for gene transfer in tissues such as the eye [38]. siRNA, a double-stranded RNA of 21–23 bp, regulates gene expression through RNA interference, which inhibits the selective degradation of the target mRNA and, ultimately, the translation of its functional protein [38,39,40]. RNA therapies, as new treatment approaches, are of great interest for us.

In this study, we worked with the Accell™ siRNA reagents, specially modified for use in difficult-to-transfect cells without transfection reagents, viruses, or electroporation [41,42]. The study aims to develop a gene therapy-based method—topical application of siRNA for knockdown of pro-apoptotic proteins—that can be used to increase the survival rate of CE of donor corneas during storage in eye banks. The focus is on preventing apoptosis of CEs. To achieve this, the basic feasibility of the project—the down-regulation of apoptosis without damaging cell viability—is to be demonstrated. Apoptosis is induced in the cells using staurosporine and etoposide to analyze the reduction of the apoptosis rate in CEs. CEs were investigated under the following two experimental conditions: (1) in vitro with an immortalized cell line and (2) in an ex vivo donor cornea. Subsequent apoptosis detection was performed by immunohistochemistry, including flow cytometry and confocal microscopy. The transfer of siRNA treatment of the donor corneas during cultivation and storage in the corneal banks should result in more donor corneal tissue being available for transplantation (Figure 1).

## 2. Materials and Methods

### 2.1. Culture Models

#### 2.1.1. In Vitro—Corneal Endothelial Cell Line

In the in vitro studies, a human corneal endothelial cell line (HCEC-12, DMSZ No. ACC-646, Braunschweig, Germany) was used [32,36,43,44]. HCECs were cultured in growth medium containing Ham’s F12/Medium 199 (1:1; Lonza, Verviers, Belgium/Sigma-Aldrich, St. Louis, MO, USA) with 5% fetal bovine serum (FBS; PAN-Biotech, Aidenbach, Germany) and 1% of 100× Antibiotic-Antimycotic (AB/AM; Thermo Fisher Scientific, Waltham, MA, USA) (complete growth medium) at 37 °C with 5% CO_2_ and passaged following standard laboratory procedures. To detach the cells, trypsin/ethylenediaminetetraacetic acid (0.05% trypsin/0.38% EDTA; Invitrogen, Gibco, Paisley, UK) was incubated at 37 °C for 5 min. This reaction was stopped with a complete growth medium, and the corresponding number of cells was used for experiments or cultivation. The experiments were achieved when cells had reached 70–80% confluence.

#### 2.1.2. Ex Vivo—Donor Cornea

In a proof-of-principle study, donor corneas with endothelium not suitable for transplantation were obtained from Cornea Bank DGFG (Deutsche Gesellschaft für Gewebetransplantation, Network for Tissue Donation, Tissue Processing, and Transplantation, Hannover and Rostock, Germany). The cornea tissue was delivered in the Culture Medium I (#P04-09701, PAN Biotech, Aidenbach, Germany) with 2.5% FBS (complete growth medium). The whole corneas were cut into four pieces/quartered for the experiments, and each quarter was placed on the endothelium layer in a well with a medium. After 96 h of incubation at 37 °C, the quarter corneas were replaced in Culture Medium II (w: Dextran 500, #P04-09702, PAN Biotech), and the tissue was investigated.

All experiments with human tissue were approved by the local ethics committee and adhered to the tenets of the Declaration of Helsinki (A 2020-0108). The investigations were carried out with four corneas. Table 1 presents donor information regarding the tissue samples used.

### 2.2. Inhibition of Pro-Apoptotic Proteins by Small Interfering RNA (siRNA)

Dharmacon™ Accell™ siRNA reagents from Horizon Discovery Ltd. (SMARTpool system; Cambridge, UK) were used for gene silencing according to the manufacturer’s recommendations. The self-delivering siRNA modification allows gene knockdown without transfection reagent or manipulation [15,41,45]. Briefly, 2 × 10^5^ HCECs were seeded onto 48-well plates or 4-well Ibidi slides (TUNEL staining) for 24 h before Accell™ siRNA application. At 70–80% cell confluence, the complete growth medium was replaced with medium without FBS (no-serum medium) and the appropriate Accell-siRNA approach was as follows: (i) untreated cells (Cells only, use of the no-serum medium only), (ii) cells with Accell Non-targeting Control pool (D-001910-10-50) (Control-siRNA), as well as (iii) cells treated with Accell Human BAX (581) siRNA-SMARTpool (E-003308-00-0050) and Accell Human BAK1 (578) siRNA-SMARTpool (E-003305-00-0050) (Bax+Bak-siRNA). The respective Accell siRNAs were dissolved in the growth medium devoid of serum (1 µM siRNA concentration).

For the ex vivo setup, one quarter of cornea was cultured in complete medium (intern control), the others with no-serum medium for the experiments under the following conditions: (i) Cells only, (ii) Control-siRNA, and (iii) Bax+Bak-siRNA. Transfection with Accell siRNA (1 µM each) of explanted human corneas was performed using the “cornea in a cup” approach, wherein the endothelial layer was facing upwards.

After 48 h, all approaches must change the medium, and the corresponding complete growth medium (with FBS) must be added. After an additional 48 h (96 h in total), the silencing was successful (Appendix A), and the viability studies (HCECs) or induction of apoptosis with subsequent apoptosis assay could be performed (Figure 2).

### 2.3. Cell Viability of HCECs

The viability of HCECs after a 96-hour transfection period was determined by MTS (3-(4,5-dimethylthiazol-2-yl)-5-(3-carboxymethoxyphenyl)-2-(4-sulfophenyl)-2H-tetrazolium; CellTiter96^®^ AQueous One Solution Cell Proliferation Assay, Promega Corporation, Madison, WI, USA) and crystal violet assay (Neisser solution II, Carl Roth GmbH + Co. KG, Karlsruhe, Germany) [46,47].

After a 96-hour transfection period, the medium must be replaced with an MTS solution (1:6 in medium). After 2 h incubation at 37 °C, 100 µL of the supernatant was taken and pipetted into a 96-well microplate (three technical replicates) to record the absorbance at λ = 492 nm using a plate reader (ELISA Reader, Anthos 2010, Biochrom, Cambridge, UK). The background measurement was carried out at 650 nm.

The MTS solution was removed entirely to quantify the optical density of adherent-growing HCECs after a 96-hour transfection period. Afterward, HCECs were washed three times with PBS and fixed using methanol. After rewashing with 0.05% Tween20/PBS (3×; VWR Chemicals, Leuven, Belgium), HCECs were stained with a 0.1% Neisser solution (Carl Roth) by 20 min shaking. After further washing steps with double-distilled water, the bound Neisser solution can be extracted by acetic acid (33%; J. T. Baker, Deventer, Netherlands). The supernatant (100 µL, three technical replicates) was transferred into a 96-well microplate, and the absorbance was measured at λ = 620 nm by the Anthos plate reader.

### 2.4. Apoptosis Induction in HCECs

After a 96-hour transfection period, apoptosis was induced in corneal endothelial cells (HCECs and donor corneas) using the topoisomerase II inhibitor etoposide (42.5 µM; Sigma-Aldrich) for 21 h. In HCECs, the apoptosis-inducing reagent staurosporine from streptomyces (250 nM; 99%; Thermo Scientific Chemicals, Loughborough, UK) was also applied for 6 h [32]. Dimethyl sulfoxide (DMSO) was added in the respective concentrations for controls as both the inducers were dissolved in DMSO.

### 2.5. Morphology of HCECs

A light microscopic evaluation was performed to monitor HCECs’ morphology after transfection and apoptosis induction. After corresponding exposure time, cell morphology was monitored by inverted bright field microscopy (Axiovert 40, Carl Zeiss AG, Jena, Germany) in a phase–contrast mode with 32× objective (LD A-Plan 32/0.40 Ph1, Carl Zeiss). The documentation was performed by an additive camera (AxioCam ICc c1, Carl Zeiss) with manual image acquisition (TIFF) using the software AxioVision (AxioVs40 64 V 4.9.1.0 for Windows, Carl Zeiss).

### 2.6. Apoptosis Assays

#### 2.6.1. Caspase-3 Activity

Measurement of apoptosis using an approach to detect the activity of a cysteine-aspartic acid protease (caspase) family member. The sequential activation of caspase plays a central role in executing cell apoptosis, with caspase-3 being activated early in apoptosis and appearing involved in the proteolysis of several important molecules.

To detect the activity of caspase-3 in HCECs after transfection and apoptosis induction (etoposide or staurosporine), cells were lysed with ice-cold lysis buffer (20 mM HEPES, 84 mM KCl, 10 mM MgCl_2_, 200 μM EDTA, 200 μM EGTA, 0.5% NP-40, 1 μg/mL leupeptin, 1 μg/mL pepstatin, and 5 μg/mL aprotinin) on ice for 10 min. Afterward, ice-cold reaction buffer (50 mM HEPES, 100 mM NaCl, 10% sucrose, 0.1% CHAPS, 2 mM CaCl2, and 13.35 mM DTT) with the fluorogenic substrate Ac (N-acetyl)-DEVD (Asp-Glu-Val-Asp)-AMC (7-amino-4-methylcoumarin) (70 µM; BD Biosciences, Heidelberg, Germany) was added and incubated at 37 °C for 1 h. The samples were transferred into a 96-well microplate, and the fluorescence intensity of the cleavage of the substrate DEVD-AMC (Ex λ = 380 nm, Em λ = 450 nm) was detected by microplate reader Fluoroskan Ascent FL (Thermo Scientific).

#### 2.6.2. TUNEL Assay

One option to count the proportional amount of apoptotic HCECs after incubation with apoptosis inducers was by using a deoxyuridine-5′-triphosphate-digoxigenin (dUTP) nick-end labeling (TUNEL) assay on a confocal laser scanning microscope (cLSM). The assay recognizes the early stage of apoptosis, characterized by DNA degradation resulting in double- and single-stranded breaks (nicks), by labeling the free 3′-OH termini. This TUNEL assay detected and quantified apoptotic DNA fragmentation in HCECs and ex vivo donor corneas.

After transfection and apoptosis induction, cells were fixed with 4% PFA, permeabilized with 0.1% Triton X-100 (each 10 min for HCECs, or 1 h for cornea tissue), and blocked with 2% BSA (bovine serum albumin solution; Sigma-Aldrich). According to the manufacturer’s instructions, TUNEL labeling was carried out with the “In Situ Cell Death Detection Kit, TMR red” (Merck KGaA, Roche, Darmstadt, Germany). Briefly, addition of TUNEL mixture (1:10) for 1 h at 37 °C. HCECs were additionally stained with Zonula occludens-1-antibody (ZO-1 Polyclonal Antibody, 1:50; Invitrogen), followed by the secondary antibody Alexa Fluor^®^488-labeled goat anti-rabbit (1:200; Life technologies by Thermo Fisher Scientific) to record the tight junctions.

To detect and count the total number of cells, 4′,6-diamidino-2-phenylindole (DAPI, nucleic acid/nucleus staining) was used at the end of the staining procedure, which was contained in the embedding medium (Fluoroshield^TM^ with DAPI, Sigma-Aldrich).

To visualize the markers in the cells served by an inverted cLSM (LSM780, Carl Zeiss AG, Oberkochen, Germany) equipped with 63× oil (63× Plan-Neofluar, 1.25 oil/0.17; Carl Zeiss) or 40× water immersion objective (C Apochromat, 1.20 W Korr M2), and the ZEN software (ZEISS efficient navigation, ZEN 2011SP4, black edition, Carl Zeiss).

TUNEL-positive HCECs were counted using ImageJ software (developed by Wayne Rasband, National Institutes of Health, Bethesda, MD, USA) and calculated to the total number of cells (Dapi-stained cells).

#### 2.6.3. Annexin V-FITC/PI

Flow cytometry with Annexin V-FITC (FITC) and propidium iodide (PI) staining was used to measure the proportion of apoptotic HCECs after apoptosis incubation. The eBioscience™ Annexin V-FITC Apoptosis Detection Kit (Thermo Fischer Scientific) is utilized to distinguish cells undergoing early or late apoptosis. Annexin V marks phosphatidylserine (PS) on the membrane’s outer surface when the cells are in the early apoptosis phase. In late apoptosis, cell integrity is lost, allowing the viability dye PI to bind to the DNA inside, and a signal for both PI and Annexin V can be analyzed.

At the end of the transfection and apoptosis induction, HCECs were washed and trypsinized. To recover cells in suspension, HCECs were tranferred into FACS tubes (BD Biosciences, Heidelberg, Germany). According to the manufacturer’s instructions, cells were resuspended in binding buffer and, after the centrifugation step, incubated with Annexin V-FITC mix in the dark at room temperature for 10 min. After further centrifugation, PI was added to the cells and incubated in the dark on ice for 15 min. The apoptosis analysis was immediately measured in the FACSCalibur (BD Biosciences) flow cytometer equipped with an argon-ion laser (488 nm). For data acquisition (Dotplot PI/FITC of at least 10,000 events), the software CellQuest Pro 4.0.1 (BD Biosciences) was used. By FlowJo_V.10.1r1 (FlowJo LLC, BD Becton Dickinson, and Company, Franklin Lakes, NK, USA) cell population was as follows: vital (FITV−/PI−), early apoptotic (FITC+/PI−), late apoptotic (FITC+/PI+), and necrotic (FITC−/PI+) were distinguished.

### 2.7. Statistical Analysis

All in vitro and ex vivo setups were conducted at least three times in independent experiments (sample size *n*). GraphPad Prism Version 7.02 (GraphPad Software Inc., La Jolla, CA, USA) was used to examine the statistical significance and graph preparation.

Data are presented in bar charts with the mean ± standard error of the mean (s.e.m.). First, normal distribution (Shapiro–Wilk normality test) and homoscedasticity (Bartlett’s test for equal variances of k samples) were done. According to the assumptions, the group differences were considered by analysis of variance, i.e., either a nonparametric Friedman test post-tested with Dunn’s multiple comparison tests (relative cell viability, caspase-3 activity) or a parametric repeated measure (RM) one-way ANOVA posthoc uncorrected Fisher’s LSD (TUNEL, Annexin V-FITC/PI) was performed. The observed differences in *p*-values with a two-sided significance level of 0.05 were deemed statistically significant.

## 3. Results

In a preliminary study, various siRNA formulations from different suppliers were tested. Of these, pooled Accell siRNAs demonstrated promising results in silencing target proteins (Bax and Bak) in HCECs. The Bak and Bax siRNA concentrations used in this study were 1 µM. Following transfection with the respective siRNAs, a significant downregulation of Bak and Bax expression was observed in HCECs over 120 h (Appendix A).

This study aimed to demonstrate the viability of transfection with Accell™ siRNA and its potential to reduce apoptosis in endothelial cells of (i) cell lines HCECs and (ii) donor corneas.

### 3.1. Unchanged Cell Viability after Accell siRNA Transfection

It is essential to analyze the transfection method’s effects on cell viability to ensure successful transfection without adverse effects. The relative cell viability (per cell) was determined using the MTS assay and crystal violet staining. Colorimetric analysis of relative cell viability showed that non-transfected (Cells only: 0.7; mean) and transfected (Control-siRNA: 0.71 and Bax+Bak-siRNA: 0.68; mean) HCECs were similarly viable, confirming that the Accell siRNA transfection procedure has no toxic or deleterious effects (Figure 3).

### 3.2. Evaluation of HCEC Morphology after Apoptosis Induction

Following the successful transfection of HCECs without any adverse effects on cell viability, the next step was to test whether the silencing of Bax and Bak proteins could prevent apoptosis when exposed to an apoptotic stimulus. Therefore, 96 h after transfection, the cells were treated with staurosporine (250 nM, 6 h) or etoposide (42.5 µM for 21 h) to induce apoptosis. In the first step, the variations in cell morphology were evaluated under a light microscope (Figure 4). The morphology of HCECs is a reliable indicator of cell vitality and can be altered in apoptotic states.

Without an apoptotic stimulus (*w*/*o* inducers, last row), the HCECs exhibited their physiological morphology, comprising a closed cluster of mosaic-like arranged hexagonal cells. Induction of apoptosis, however, resulted in the control cells (Cells only and Control-siRNA) having a pathological phenotype, i.e., damaged cells that had lost their characteristic hexagonal cell shape and could no longer form a closed cell cluster (numerous gaps, Figure 4 arrows). The morphological changes associated with apoptosis were observed under staurosporine treatment (Figure 4, first row). The stability of the cells was abolished, and the cells shrank, preventing the formation of cell clusters.

In contrast, HCECs transfected with Bax+Bak-siRNA preserved the typical morphology of hexagonal cells and formed dense cell-cell contacts of neighboring cells. Based on the morphological assessment, it was evident that apoptosis could be prevented in HCECs by adding Bax+Bak-siRNA.

### 3.3. Bax+Bak-siRNA Transfection Reduced Caspase-3 Activity in HCECs

Given the similarity in morphology between HCECs treated with Bax+Bak-siRNA and vital cells, we sought to quantify the reduction in apoptosis. For this purpose, the activity of caspase-3, an essential trigger of apoptosis, was determined on the fluorometer. As illustrated in Figure 5a,b, Bax+Bak-siRNA treatment significantly reduced caspase-3 activity in HCECs compared to controls (Cells only and Control-siRNA). HCECs that were transfected with Bax+Bak-siRNAs and treated with staurosporine demonstrated a significant reduction in caspase-3 activity compared to controls (to Cells only: 59.3%; to Control-siRNA: 65.4%; Figure 5a). When etoposide was employed to induce apoptosis, there was also a significant reduction in caspase-3 activity in HCECs transfected with Bax-Bak compared to controls (to Cells only: 51%; to Control-siRNA: 58.5%; Figure 5b). Consequently, it was demonstrated that the silencing of pro-apoptotic proteins Bax-Bak resulted in a reduction in cell death (after intrinsic and extrinsic apoptosis induction with etoposide and staurosporine), thereby conferring a significant degree of protection on HCECs.

### 3.4. Inhibition of Pro-Apoptotic Proteins Results in Reduced Apoptosis Rates

After apoptosis induction in transfected HCECs (staurosporine or etoposide), apoptotic cells with DNA fragmentation were visualized by deoxyuridine-5′-triphosphate-digoxigenin (dUTP) nick-end labeling (TUNEL) and examined by confocal laser scanning microscopy. A marker of the endothelial cells—ZO1—was also visualized to indicate the tight junction formation (cell-cell contacts).

Figure 6 displays the results of the TUNEL assay. The exemplary fluorescence microscopic images show an apparent effect of apoptosis (Figure 6a). Without apoptosis inducers (Figure 6a, last row), endothelial cells can be detected with few DNA fragmentations (TUNEL-positive cells in red). Even in the Bax+Bak-siRNA transfected cells, only a few apoptotic nuclei can be found after apoptosis stimulus. In comparison, after the stimulus, many TUNEL-positive cells were detectable in the control cells (Cells only and Control-siRNA). Other signs of ongoing apoptosis, the loss of tight junction integrity, were also visible. Due to the detection of the tight junction protein ZO1 (green) formation, which marks the cell border between neighboring cells, the loss of cell-cell contact could be detected in the control cells after apoptosis stimulus. Intercellular openings between the neighboring cells after etoposide treatment (Figure 6a, middle row) or the complete loss of the tight junction under staurosporine (Figure 6a, first row) were detectable in the controls. In contrast, a clear continuous ZO-1 band was found in the Bax+Bak-siRNA transfected cells after treatment with etoposide and in the cells without apoptosis induction, indicating an intact vital cell layer. The cell junction protection of Bax+Bak-siRNA was less evident for staurosporin.

Results from these experiments demonstrate that silencing pro-apoptotic proteins Bax and Bak by Accell™ siRNA leads to a significant decrease in apoptosis relative to the controls (Cells only, Control-siRNA) (Figure 6b).

Cells without apoptosis stimulus showed an 11% apoptosis rate (Cells only: 11.5 ± 4.2%, Control-siRNA: 15.9 ± 5.4%, Bax+Bak-siRNA: 7.1 ± 2.3%; mean ± s.e.m.). After apoptosis induction, the number of TUNEL-positive cells increased significantly in the controls: Cells Only (staurosporine: 41.7%, etoposide: 32.8%) and Control-siRNA (staurosporine: 41.5%, etoposide: 51.6%) compared to the Bax+Bak-siRNA transfected cells (staurosporine: 13.2%, etoposide: 7.5%).

### 3.5. Bax+BaksiRNA Transfection Results in Reduced Development of Apoptosis

In the next step, the phases of the apoptotic cell populations after apoptosis stimulus (staurosporine or etoposide) were quantified using Annexin V-FITC and propidium iodide (PI) staining on the flow cytometer. The number of vital living cells was significantly higher in the Bax+Bak-siRNA transfected HCECs, independent of the apoptosis-inducing reagent (87%, Figure 7).

After staurosporine treatment, the percentage of early and late apoptotic cells in all control cells (Cells only and Control-siRNA, ~20%) was significantly increased compared to cells after Bax+Bax-siRNA transfection (~6%) (Figure 7a). After apoptosis induction with etoposide, it was found that the percentage of early and late apoptotic cells was significantly increased only in untransfected cells (Cells only) compared to Bax+Bax-siRNA-treated cells. Furthermore, the cells were more detectable in the early phase of apoptosis (Figure 7b). In contrast, we found an increased proportion of control cells in the late apoptosis phase upon staurosporine induction, which is consistent with the microscopic images, as staurosporine treatment always resulted in severe shrinkage (Figure 4, top row) and loss of cell-cell contacts in the control cells (Figure 6a, top row).

Without inducers, we could observe 75% of living cells, 3% necrotic cells, 12% early apoptotic cells, and 10% late apoptotic cells. Based on the flow cytometry data, it can be concluded that the silencing of pro-apoptotic proteins Bax and Bak causes reduced HCEC apoptosis.

### 3.6. Reduced Apoptosis Rate in Human Donor Corneal Endothelial Cells

In order to test the hypothesis that pro-apoptotic gene transfer can promote corneal endothelial survival in an eye bank environment, the following experiment was used to analyze the apoptosis rate in donor corneas. The donor corneas were quartered to assess apoptosis, i.e., one-quarter was untransfected (Cells only), the second transfected with Control-siRNA, and the third with Bax+Bak-siRNA. They were subjected to the same conditions as the cell line, and apoptosis was induced with etoposide after 96 h. The rate of apoptotic cells was quantified using the TUNEL assay on a confocal microscope.

In the ex vivo evaluation of donor corneas cultured under the control conditions (Cells only and Control-siRNA), numerous TUNEL-positive cells were visible under the microscope (TUNEL—red, Figure 8a). In comparison, less DNA fragmentation (TUNEL-positive cells) could be detected in the corneal endothelia of donor corneas transfected with Bax+Bak-siRNA. Quantification of three corneas showed a significant inhibition of apoptosis in corneal endothelia transfected with Bax+Bak-siRNA (Figure 8b). Thus, the results showed that Accell Bax+Bak-siRNA also reduced apoptosis in donor cornea tissue after 96 h.

## 4. Discussion

Corneal transplantation is the only therapeutic option to permanently improve vision for many blind people, especially for younger people of working age [13,14,48]. However, corneal endothelial (CE) cells undergo apoptosis, programmed cell death, during the cultivation of donor corneas [17,24,32,35,36,49,50]. In general, transplanted organs are confronted with various immunological and metabolic stress factors that increase the proportion of apoptotic cells [51,52]. Given the current lack of transplants, it is of great interest to improve the quality of transplanted organs. Organ preservation represents an unutilized therapeutic window with great potential to improve graft quality [51,52]. Various gene- and cell-based therapeutic approaches in the human eye (retina and CE) have been presented to achieve more prolonged survival or higher resistance to stress [17,24,25,26,27,28,35,36,38,49,53,54,55,56]. Based on the large amount of literature described, it is already clear that no optimal therapy has yet been established [26]. Overall, gene therapy is still behind progress, as the molecular and biochemical complexities of apoptosis are not yet fully understood [28,30].

The following two types of therapeutic approaches have been described that target CEs: (1) support of proliferation [57] and (2) suppression of apoptosis [17,24,36], which is the strategy used in the study. Since CEs do not typically proliferate [3,4], and while promoting proliferation carries the risk of triggering certain pathologies, suppression of CE apoptosis prevents tumor development [35]. Studies showed that cells can be rescued from the apoptotic program when the apoptotic stimulus is removed [17,24,30,34,35,36,58,59]. For example, inhibition of apoptosis protease activating factor-1 (Apaf-1), a key regulatory enzyme of the apoptosis pathway, by a novel inhibitor (ZYZ-488) was found to have a protective effect on cardiomyocytes in cardiac ischemia [60].

Previous studies have successfully presented approaches with the overexpression of anti-apoptotic or survival-promoting proteins such as BCL-XL or p35 in CE [17,24,34,35,36]. However, overexpression of proteins leads to side effects such as an “overload” of the cell with the respective protein function and extensively utilized protein machinery, leading to unbalanced protein production [28,61]. While the successful overexpression of anti-apoptotic proteins has already been described, the application of silencing of pro-apoptotic proteins in CE is described for the first time in this study. Selective knockdown of pro-apoptotic proteins such as Bax (BCL2-associated X, apoptosis regulator) and Bak (BCL2 antagonist/killer 1) is a smooth way to shift the apoptotic balance in cells and prevent apoptosis [27,31]. Studies demonstrate that the downregulation of pro-apoptotic Bax and Bak proteins reduces apoptosis in glioma cells (C6) and human proximal tubule epithelial cells (HK-2), respectively [62,63,64]. This is the first report of a siRNA-based approach to prevent corneal endothelial cell apoptosis in a translational setting. Gene silencing induced by small interfering RNA (siRNA) has been successfully described for treating ocular pathologies such as corneal neovascularization, dry eye, or glaucoma [41].

It is essential to choose a transfection reagent that does not limit cell viability while maintaining a desirable transfection rate [17,27,28,61,65,66,67,68,69]. Studies have shown that Accell™ siRNA, with a self-releasing siRNA modification, enables gene knockdown without transfection reagent or manipulation [45,70]. Accell™ siRNA has already been successfully used in hematopoietic stem and progenitor cells and is described as transient, reversible, and dose-variable [41].

To determine the biocompatibility of this transfection method, we analyzed the metabolic activity of HCECs after Accell siRNA transfection (96 h after addition). In vitro, all tested Accell™ siRNA (Pool non-target, Pool Bax, Pool Bak) showed no impairment of cell viability, so we could continue using the Accell™ siRNA without concern. In order to test the efficiency of the Accell™ siRNA, apoptosis was induced in the cells after the transfection procedure, and the apoptosis rate was subsequently analyzed using various assays. Numerous ways exist to induce, control, and inhibit cell apoptosis [30,71,72]. In this study, staurosporine, a potent protein kinase C PKC inhibitor that induces apoptosis via caspase-3 [73,74], as well as etoposide, a topoisomerase II inhibitor that causes permanent double-strand breaks, were used [32,75]. Etoposides can act via the “intrinsic or mitochondrial” cell death pathway or the “extrinsic pathway—death receptor” [75].

The complete in vitro setup was documented by light microscopy. After the 96-hour transfection with the Accell-siRNA, typical hexagonal cells with a dense mosaic cell structure were visible. After adding staurosporine and etoposide, the loss of contact with the neighboring cell or ECM could be detected in the control cells (Cells only, Control-siRNA). By inducing apoptosis with staurosporine, not only cell retraction, the typical cell shrinkage with densely packed organelles, but also membrane blebbing and pyknosis (condensation) could be observed in Cells only and Control-siRNA [29,30,71]. Thuret et al. [73] also observed significant morphological changes, such as cell shrinkage or membrane blebbing and loss of cell-cell contact, in HCEC after adding staurosporine. However, the treatment with Bax+Bak-siRNA had a positive effect on the survival of HCECs after the addition of apoptotic inducers (staurosporine and etoposide); a cell cluster with hexagonal HCEC was recognizable.

Apoptosis is further characterized by biochemical features in which caspase activation plays a central role [30]. It has been described that the pro-apoptotic enzymes Bax and Bak, critical players in the mitochondrial apoptosis pathway, release cytochrome c into the cytosol, which then binds to Apaf-1 and initiator caspase-9 (apoptosome), leading to the activation of effector caspases-3, -6, and -7 [34,71,76,77,78,79,80]. Intrinsic and extrinsic apoptosis pathways converge at the level of effector caspases-3 and -7 [30,72,81,82]. Caspase-3 is considered a crucial trigger of apoptosis and is responsible for the cleavage of proteins [30]. Studies on apoptosis in CE have shown that staurosporine-treated cells had increased caspase-3 activity [32,37]. Our study demonstrated that the extent of caspase-3 activation after apoptosis stimulus was significantly lower in cells than in control cells due to silencing of Bax-Bak. Rudel et al. [83] showed that caspase activity inhibition is essential in preventing apoptosis. Thus, we have the first indication of a reduced apoptosis rate after the knockdown of pro-apoptotic proteins.

Nevertheless, caspase activation is not necessarily an indication of the apoptosis process [30]. Therefore, other assays such as TUNEL (Terminal dUTP Nick End-Labeling) were used by enzymatically end-labeling the DNA strand breaks [30,84]. By detecting DNA fragmentation, we could show that silencing of Bax+Bak after treatment with etoposide and staurosporine resulted in a significantly reduced number of TUNEL-positive cells compared to control cells.

In addition to TUNEL, immunostaining of the zonula occludens protein (ZO-1), located in the submembranous region of the tight junctions [85], was also performed. The intact corneal endothelial layer characteristically forms cell-cell contacts with intercellular tight and gap junctions [2,48]. In our study, cells without apoptosis inducers and the Bax+Bak-siRNA-transfection showed a continuous ZO-1 localization at the border, indicating that the neighboring cells are closely connected. However, in the control cells (Cells only and Control-siRNA), a barrier disruption could be observed after the apoptosis stimulus, leading to large openings in the cell-cell contact zone. Hoentsch et al. [85] also demonstrated a complete loss of cell-cell contacts in dying epithelial cells after physical plasma treatment. Taylor et al. [71] described that caspase-dependent degradation of cell-cell adhesion complexes is evident in early apoptotic cells, which is one explanation for the loss of ZO-1 in control approaches with many TUNEL-positive cells. This microscopic observation supports our caspase results, which show that the number of cells undergoing apoptosis was reduced in cells transfected with Bax and Bak.

However, the TUNEL method has limitations, such as unspecific or false positive results from necrotic cells [86,87]. Therefore, this efficiency study was combined with another assay to confirm that cell death occurs by apoptosis. Biochemical features of apoptosis include caspase-dependent protein cleavage, DNA fragmentation, and the appearance of phosphatidylserine on the cell membrane surface [30,71,79,88]. During apoptosis, phosphatidylserine is translocated from the inner to the outer plasma membrane layer, which leads to early phagocytic recognition and finally induces phagocytosis by neighboring cells [30]. In the further apoptosis assay, Annexin V, a recombinant phosphatidylserine-binding protein, was therefore detected by flow cytometry. Using another dye, propidium iodide (PI), the membrane integrity can also be determined, and thus, a distinction is made between the early and late apoptotic phases [88,89]. Using this assay, we demonstrated a significant decrease in both early and late apoptotic phases when HCECs were treated with Bax+Bak-siRNA [90]. All apoptosis assays clearly showed that the knockdown of Bax and Bak leads to resistance to apoptosis induction by various apoptotic stimuli.

It has been described that the immortalized cell line HCEC-12 does not closely represent the behavior of ex vivo donor corneal endothelium [36]. Nevertheless, they represent a valid in vitro CE model to investigate cellular transfection efficiency and viability initially. Due to its translational relevance, the concept of siRNA treatment was also performed in the target tissue, ex vivo donor corneas. To bridge the gap between basic research and tissue storage, all data were obtained in full-thickness research corneas, the standard for corneal graft storage [20]. The initial results in terms of corneal tissue transfection were encouraging, as Accell™ siRNA achieved a reduction in Bax and Bak expression of approximately 25% (Appendix A). Even after apoptosis induction with etoposide, a reduced number of cells with double-strand breaks could be detected by TUNEL assay when transfected with Bax+Bak-siRNA.

For the ex vivo setup, we need donor corneas with intact endothelium. Procuring such high-quality tissue for experiments is a significant challenge, as the patient must be the first beneficiary of this tissue for ethical reasons. Therefore, unfortunately, only a few analyses could be performed on a small scale (*n* = 3). In order to test the hypothesis in corneal tissue as to whether the elimination of Bax and Bak contributes to the reduction of apoptosis, further analyses should, therefore, follow soon. Furthermore, the handling of the ex vivo corneas is a critical factor, as they are cultured in smaller volumes for 4 days and thus not stored according to eye bank protocols. However, long-term studies have shown that the damaged endothelium undergoes apoptosis after two to three weeks [23,91]. In addition, cutting/quartering the cornea opens the stroma and epithelium to the culture medium, which leads to swelling of the stroma and increases the loss of endothelial cells. However, we are grateful to receive ex vivo corneas for our studies from the DGFG and thus have the opportunity to conduct our apoptosis studies to the best of our ability. Since we can test the extent of the treatment directly in the target tissue and the initial situation (during storage in the eye bank), we are not dependent on animal experiments.

In this study, we demonstrated in the proof-of-principle that a pro-apoptotic gene transfer prevents apoptosis of CE. In a new study, we would like to further analyze the silencing of pro-apoptotic proteins and transfer siRNA into CE using a recordable, defined transfection method to provide more cost-effective alternatives to Accell™ siRNA and optimize the delivery of siRNA. Furthermore, the analysis should focus on the interaction of other signaling pathways, such as nuclear factor 2 (erythroid-derived 2) (Nrf2, antioxidant response [92,93]), in mediating the control of apoptosis and endothelial cell recovery to understand the complex interplay. Most gene therapies are primarily experimental in order to better understand the mechanisms for developing effective treatments [6].

## 5. Conclusions

The study presented here shows a clear benefit for the survival of corneal endothelial cells (CE) when pro-apoptotic proteins Bax and Bak are downregulated by siRNA. The Accell™ siRNA used was an effective strategy to exert the Bax+Bak silencing function without detectable deleterious effects in HCEC. This knockdown also leads to reduced apoptosis in ex vivo donor corneas. Applying this gene therapy approach in the cornea bank could lead to more donor corneas being available for transplantation and a reduction in graft failures. Since the number of corneal donations in Germany has been too low for decades, the results of these experiments could point the way to more effective handling of donated corneas.

## Figures and Tables

**Figure 1 biomedicines-12-01439-f001:**
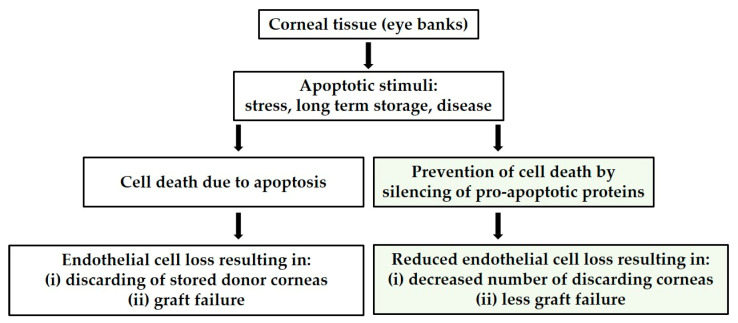
Chart illustrating the causes of endothelial cell loss leading to tissue wastage and the proposed strategy to circumvent this wastage.

**Figure 2 biomedicines-12-01439-f002:**
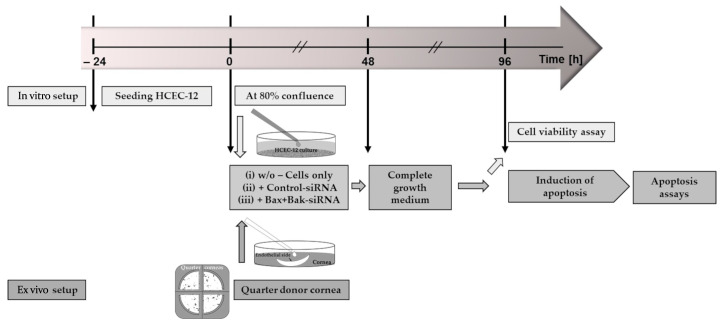
Schematic representation of the time course of the experiments. For the in vitro approach, corneal endothelial cells (HCECs) were seeded for 24 h at a density of 2 × 10^5^ to reach 80% confluence for the experiment. For the ex vivo approach, the donor cornea was quartered, and one quarter was placed with the endothelial side up in a well (“cornea in a cup”). The procedure for both cultures (highlighted in green) was then to apply siRNA in serum-free medium at time 0 h with the following treatment: (i) no addition of siRNA as control (Cells only); (ii) addition of non-target siRNA (Control-siRNA); or addition of siRNA to knockdown the pro-apoptotic proteins Bax and Bak (Bax+Bak-siRNA). After 48 h, the medium in both cultures was changed, and the appropriate complete growth medium was added. After a further 48 h of knockdown, experiments were performed for the HCEC viability assay. In order to analyze the transfection, cell death was induced with etoposides (HCECs additional with staurosporine), followed by apoptosis assays such as TUNEL.

**Figure 3 biomedicines-12-01439-f003:**
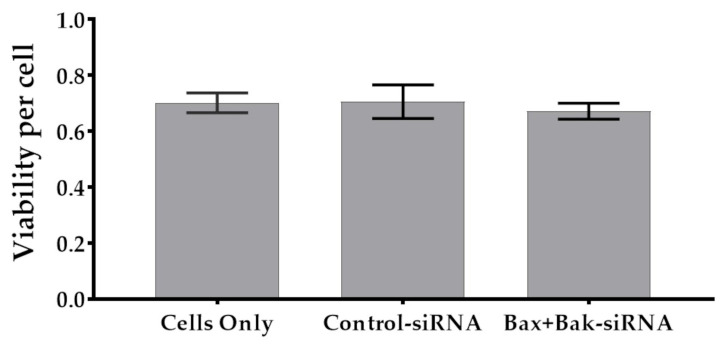
Relative cell viability of corneal endothelial cells (HCECs) after 96-hour transfection procedure. Note that the transfection with Accell siRNA showed no effect on cell viability. Cell metabolism (MTS assay) values were related to cell number (crystal violet). (Anthos reader; *n* = 4, mean ± s.e.m., Friedmann test posthoc Dunn’s multiple comparison test: not significant).

**Figure 4 biomedicines-12-01439-f004:**
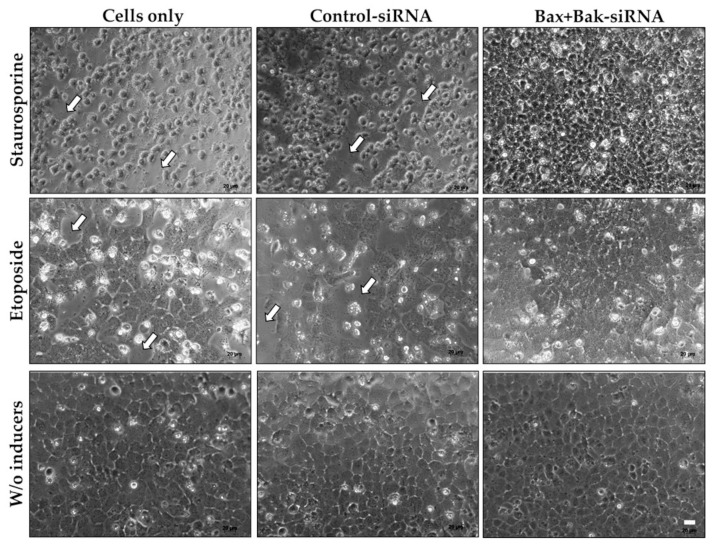
Cell morphology of human corneal endothelial cells (HCECs) under different conditions: cells were transfected with the Accell siRNA (Control-siRNA, Bax+Bak-siRNA) or untransfected (Cells only) for 96 h (*w*/*o* inducers, last row), followed by staurosporine (250 nM for 6 h; first row) or by etoposide treatment (42.5 µM for 21 h; middle row). It can be seen that the induction of apoptosis with both staurosporine and etoposide alters the phenotype of HCECs in controls (Cells only and Control-siRNA). Under staurosporine treatment, the cells shrink significantly and can no longer form cell clusters. The loss of cells and contacts can also be observed with etoposide treatment (arrows). In contrast, Bax+Bak-siRNA transfected HCECs and treated with apoptosis inducers have improved morphology, resembling cells without apoptosis stimulus (*w*/*o* inducers). (Axiovert 40C, Zeiss; bars 20 µm).

**Figure 5 biomedicines-12-01439-f005:**
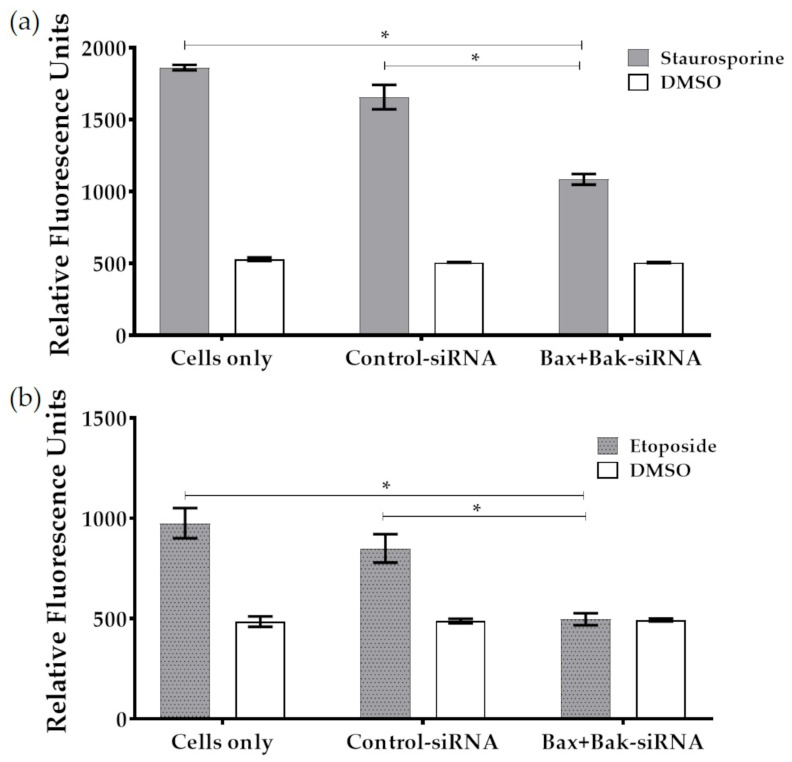
The extent of apoptosis was measured in terms of caspase-3 activity after induction of apoptosis in transfected (Control-siRNA, Bax+Bak-siRNA) and untransfected (Cells only) HCECs: (**a**) Apoptosis was induced using Staurosporine (250 nM for 6 h) after 96 h transfection. (**b**) Apoptosis was induced using etoposide (42.5 µM for 21 h) after 96 h transfection. Note that in both cases, the apoptosis rate was significantly reduced due to the knockdown of pro-apoptotic proteins Bax and Bak. (microplate reader Fluoroskan Ascent FL; *n* = 3, mean ± s.e.m., Friedman-Test posthoc Dunn’s multiple comparison test: * *p* < 0.05).

**Figure 6 biomedicines-12-01439-f006:**
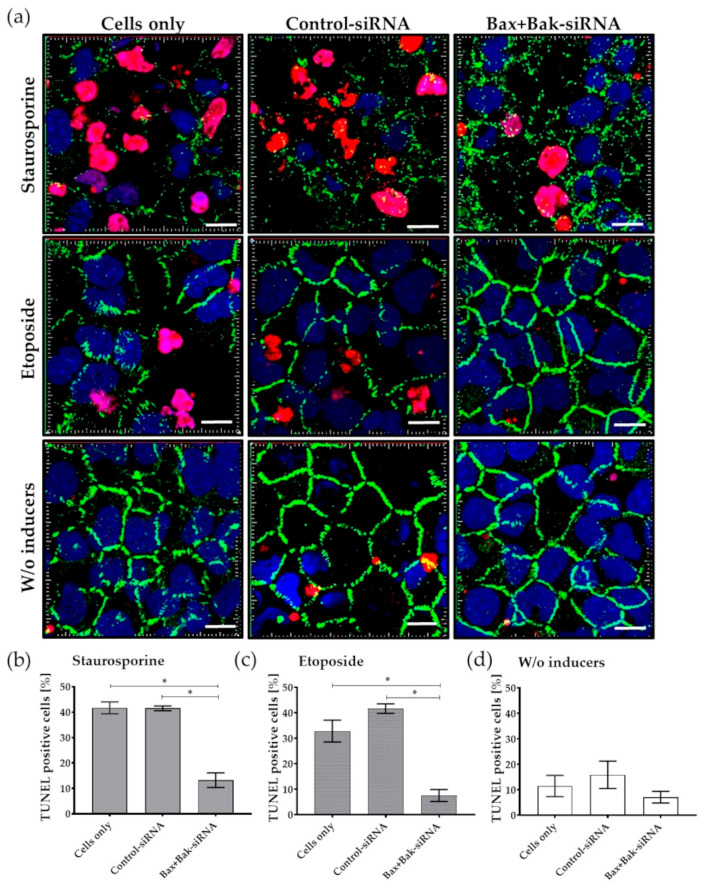
Detection of apoptosis rate by TUNEL assay in human corneal endothelial cells (HCECs) after transfection with Accell Control-siRNA or Bax+Bak-siRNA compared to untransfected cells (Cells only). After incubation for 96 h, the apoptosis inducers staurosporine (250 nM for 6 h) or etoposide (42.5 µM for 21 h) were added. (**a**) Exemplary images of apoptosis rate and changes in cell morphology in HCECs labeled with TUNEL assay and staining of cell-cell contacts (ZO-1). Note that due to the induction of apoptosis, many nuclei of apoptotic cells (red) and gaps in the cell-cell barrier (green) could be detected in controls—Cells only and Control-siRNA. In contrast, almost no TUNEL-positive cells are found in Bax+Bak-siRNA-transfected cells, and tight cell-cell contact of HCECs can be observed, comparable to endothelial cells without (*w*/*o*) apoptosis induction for etoposide and to a lesser extent for staurosporine. (LSM780, Zeiss; red: TUNEL—apoptotic DNA fragmentation, blue: Dapi—cell nuclei, green: ZO-1—cell-cell contacts; 63× oil objective, zoom 2, 3D overlay, bar = 10 µm). Images were taken from five random fields, TUNEL-positive cells were counted, and the percentage for (**b**) staurosporine, (**c**) etoposide, and (**d**) cells treated without inducers was calculated. Quantification clearly shows that the apoptosis rate was reduced after the knockdown of Bax+Bak. (ImageJ; *n* = 3, mean ± s.e.m., RM one-way ANOVA posthoc Bonferroni’s multiple comparison test: * *p* < 0.05).

**Figure 7 biomedicines-12-01439-f007:**
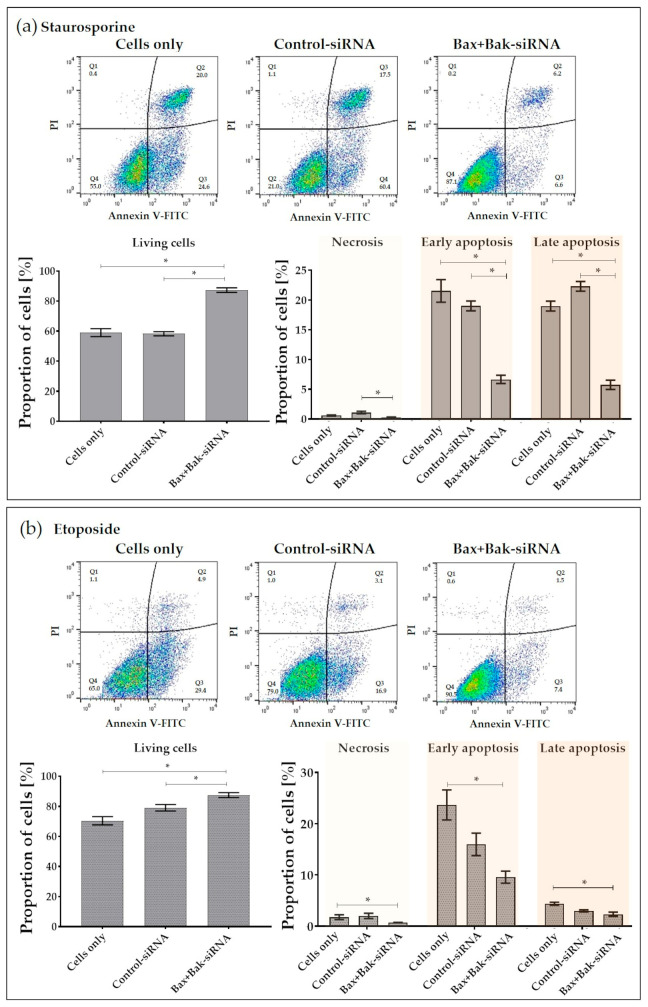
Detection of apoptosis rate using Annexin V-FITC/Propidium Iodide (PI) in human corneal endothelial cells (HCECs) after transfection with Accell Control-siRNA or Bax+Bak-siRNA compared to untransfected cells (Cells only). After incubation for 96 h, cells were incubated with either apoptosis inducers (**a**) staurosporine (250 nM for 6 h) or (**b**) etoposide (42.5 µM for 21 h). Representative flow cytometric analysis of HCECs under the respective conditions. (FACSCalibur, BD; cells in Q1: necrosis, Q2: late apoptosis, Q3: early apoptosis; Q4: live; FL-1: Annexin, FL-2: PI). Quantification shows that significantly more live cells were present after the Bax+Bak knockdown. (FlowJo; *n* = 3, mean ± s.e.m., RM one-way ANOVA posthoc uncorrected Fisher’s LSD test: * *p* < 0.05).

**Figure 8 biomedicines-12-01439-f008:**
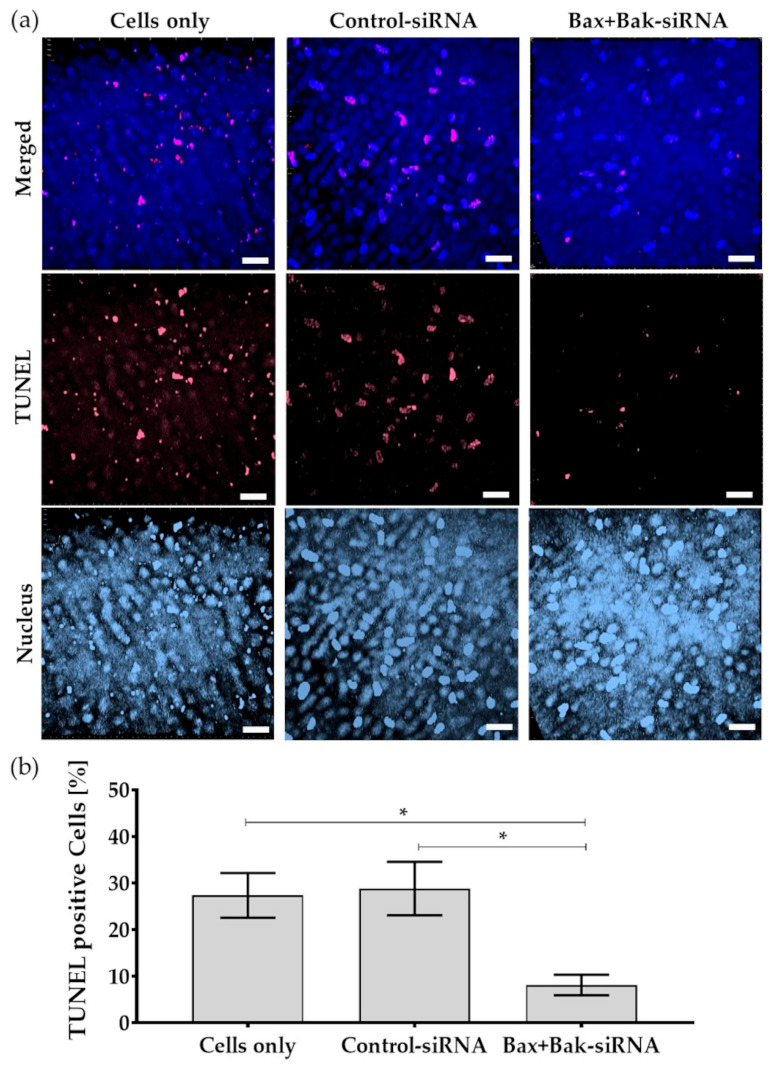
As a proof-of-principle, apoptosis was detected by TUNEL (TdT-mediated dUTP-biotin nick end labeling) assay in the endothelium of donor corneas after transfection with Accell Control-siRNA or Bax+Bak-siRNA compared to untransfected cells (Cells only). After incubation for 96 h, the apoptosis inducer etoposide (42.5 µM for 21 h) was added. (**a**) Exemplary images of the nuclei of the cells labeled with TUNEL (red) and Dapi (blue). Note that despite the induction of apoptosis, fewer nuclei of apoptotic cells were visible in the corneal endothelium compared to controls—Cells only and Control-siRNA. (LSM780, Zeiss; red: TUNEL—apoptotic DNA fragmentation; blue: Dapi—cell nuclei; 40× objective, zoom 0.6, 3D overlay, bar = 40 µm). (**b**) Quantification indicated that the apoptosis rate was significantly reduced in corneas after the Bax+Bak knockdown. (ImageJ; *n* = 3, mean ± s.e.m., RM one-way ANOVA posthoc uncorrected Fisher’s LSD test: * *p* < 0.05).

**Table 1 biomedicines-12-01439-t001:** Donor information.

Factor	Donor
Average age, y *	74 ± 7
Sex, female/male	2/2
Average cell density [cells/mm^2^] *	1785 ± 235

* Mean ± std. deviation.

## Data Availability

The data sets generated during and/or analyzed during the current study are available from the corresponding author on reasonable request. The data set is stored on the local Rostock University Medical Center (UMR) server.

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
