# Peer review of "Suppressing Pro-Apoptotic Proteins by siRNA in Corneal Endothelial Cells Protects against Cell Death"

_biomedicines, 2024, doi:10.3390/biomedicines12071439_

Round 1
Reviewer 1 Report
Comments and Suggestions for Authors
Summary:
This is an excellent study in which the role of eye bank storage dependent rises in apoptosis is assessed in mediating time dependent declines in corneal endothelial viability that reduce restoration of normal transparency following a corneal transplant. To make this evaluation, the effects are measured of small interfering (si)RNA-mediated silencing of BAX and BAK gene expression on reducing declines in corneal endothelial survival induced by increasing apoptosis with either etoposide or staurosporine. Transfection with relevant Bax and Bak siRNA suppressed BAX and BAK gene expression levels in the corneal endothelial cell line (HCEC-12) and they reduced rises in apoptosis. In Bax+Bak-siRNA transfected HCECs the cell death rate was lower in the Bax+Bak-siRNA transfected group than in the control group transfected instead with irrelevant siRNAs. The TUNEL assay results showed that in the in vivo donor Bax+Bak-siRNA corneas, the TUNEL-positive cells are fewer than in the control-treated corneas. These results suggest that downregulating BAK and BAX gene expression levels may improve the outcome of eye bank storage since inhibition of apoptosis may prolong preservation of corneal endothelial function and viability. These differences suggest that BAX and BAK downregulation may increase the likelihood that transplantation surgery results in longer lasting restoration of normal vision. The impact of this innovative study will be improved if the interrelationship is determined between changes in BAX and BAK and Nrf2 gene expression. In addition, there are two minor concerns that are listed below.
Major Concern:
The manipulation and handling of corneal tissue prior to and during performance of transplantation surgery can result in tissue injury. Such stress can promote increases in apoptosis through activating the same signaling pathway that oxidative stress stimulates.
Activation of Nrf2 is implicated as a potential protective factor in many ocular diseases involving oxidative stress and inflammation, including ocular surface disease, keratoconus, dry eye, cataract, glaucoma, uveitis, diabetic retinopathy, and age-related macular degeneration [PMID 32273949] Nrf2 activation has also experimentally been shown to modulate neuroinflammation and neurodegeneration [ PMID 23942037].In the eye, Nrf2 has been tested as a potential therapeutic target preventing ocular diseases including dry eye, cataract, uveitis, and various retinopathies [ PMID28473877].
These considerations prompt determining if activation of Nrf2 gene expression by rises in BAK and BAX gene expression contributes to maintenance of endothelial cell function during tissue storage and manipulation prior to surgery. It is relevant to determine if selective inhibition of Nrf2 activation with ML385 alters the effects of BAX and BAK miRNA transfection on apoptosis induced by staurosporine or etoposide [ PMID 363305267]. If inhibition of Nrf2 activation alters the recovery of endothelial vitality and function induced by oxidative stress, this result suggests that an interrelationship exists between changes in BAK and Nrf2 expression in mediating control of apoptosis and endothelial cell recovery.
Minor Concerns:
Line 26: change preventing CE to inhibiting
Line 443: change will to was used
Author Response
Reviewer 1:
Biomedicines-3058990
Suppressing pro-apoptotic proteins by siRNA in corneal endothelial cells protects against cell death
Staehlke et al.
Comments and Suggestions for Authors
Summary:
This is an excellent study in which the role of eye bank storage dependent rises in apoptosis is assessed in mediating time dependent declines in corneal endothelial viability that reduce restoration of normal transparency following a corneal transplant. To make this evaluation, the effects are measured of small interfering (si)RNA-mediated silencing of BAX and BAK gene expression on reducing declines in corneal endothelial survival induced by increasing apoptosis with either etoposide or staurosporine. Transfection with relevant Bax and Bak siRNA suppressed BAX and BAK gene expression levels in the corneal endothelial cell line (HCEC-12) and they reduced rises in apoptosis. In Bax+Bak-siRNA transfected HCECs the cell death rate was lower in the Bax+Bak-siRNA transfected group than in the control group transfected instead with irrelevant siRNAs. The TUNEL assay results showed that in the in vivo donor Bax+Bak-siRNA corneas, the TUNEL-positive cells are fewer than in the control-treated corneas. These results suggest that downregulating BAK and BAX gene expression levels may improve the outcome of eye bank storage since inhibition of apoptosis may prolong preservation of corneal endothelial function and viability. These differences suggest that BAX and BAK downregulation may increase the likelihood that transplantation surgery results in longer lasting restoration of normal vision. The impact of this innovative study will be improved if the interrelationship is determined between changes in BAX and BAK and Nrf2 gene expression. In addition, there are two minor concerns that are listed below.
Reviewer1: Major Concern:
The manipulation and handling of corneal tissue prior to and during performance of transplantation surgery can result in tissue injury. Such stress can promote increases in apoptosis through activating the same signaling pathway that oxidative stress stimulates.
Activation of Nrf2 is implicated as a potential protective factor in many ocular diseases involving oxidative stress and inflammation, including ocular surface disease, keratoconus, dry eye, cataract, glaucoma, uveitis, diabetic retinopathy, and age-related macular degeneration [PMID 32273949] Nrf2 activation has also experimentally been shown to modulate neuroinflammation and neurodegeneration [ PMID 23942037].In the eye, Nrf2 has been tested as a potential therapeutic target preventing ocular diseases including dry eye, cataract, uveitis, and various retinopathies [ PMID28473877].
These considerations prompt determining if activation of Nrf2 gene expression by rises in BAK and BAX gene expression contributes to maintenance of endothelial cell function during tissue storage and manipulation prior to surgery. It is relevant to determine if selective inhibition of Nrf2 activation with ML385 alters the effects of BAX and BAK miRNA transfection on apoptosis induced by staurosporine or etoposide [ PMID 363305267]. If inhibition of Nrf2 activation alters the recovery of endothelial vitality and function induced by oxidative stress, this result suggests that an interrelationship exists between changes in BAK and Nrf2 expression in mediating control of apoptosis and endothelial cell recovery.
- Authors: Authors: In this first study, we investigated the potential of gene therapy with anti-apoptotic siRNAs in donor corneas to prolong survival. The study is essential due to the worldwide shortage of high-quality corneas for transplantation.
Thanks for your suggestion and excellent advice. We will also look at the interaction between the changes in Bax, Bak, and Nrf2 gene expression in the future. In future projects, for example, we will also use microarrays to analyze the gene expression of the oxidative stress signaling pathway and thus nuclear factor- (erythroid-derived 2-) like 2 (Nrf2), another critical aspect of apoptosis of donor corneas during storage. The interaction of the various signaling pathways in the cell, such as the apoptosis cascade and Nrf2, should be investigated in more detail.
Due to its structure and function, the cornea is exposed to mechanical stress, which is the source of oxidative stress and, thus, reactive oxygen species (ROS) [1]. Nrf2-mediated antioxidant defense is key in regulating apoptosis [1-4]. Thanks to your literature, we got a good overview of Nrf2 signaling [1-3].
Nrf2, a regulator of many life processes, is essential in antioxidant, anti-inflammatory and anti-fibrotic reactions [1,2]. In many corneal diseases such as cataracts, glaucoma, uveitis, and AMD, increased ROS can lead to damage to DNA, lipids, and proteins. Signaling pathways of Nrf2 have been shown in various studies to regulate oxidative stress [1-4]. Therefore, Nrf2 activation is also a potential target that protects the CE from stress [2]. Using Nrf2 activators or overexpression of Nrf2 can reduce DNA fragility by promoting cytoplasmic stability and final translocation of Nrf2 [1-2]. Since Nrf2 acts as an essential link between cell survival and antioxidant gene expression, activation of Nrf2 can promote viability and reduce apoptosis [1-2]. One study demonstrated that the protective effect of inhibiting oxidative stress, inflammation, and apoptosis is exerted by activating Nrf2 and inhibiting NF-kB and Caspase/Bax signaling pathways [4].
Unfortunately, we can no longer perform such investigations within the scope of this study, but for future projects, analyzing the interrelationship between changes in Bax, Bak, and Nrf2 expression in mediating the control of apoptosis and endothelial cell recovery is an important focus. Underlying molecular mechanisms should be further analyzed to facilitate the development of new therapies.
[1] Wang MX, Zhao J, Zhang H, et al. Potential Protective and Therapeutic Roles of the Nrf2 Pathway in Ocular Diseases: An Update. Oxid Med Cell Longev. 2020;2020:9410952. doi:10.1155/2020/9410952
[2] Batliwala S, Xavier C, Liu Y, Wu H, Pang IH. Involvement of Nrf2 in Ocular Diseases. Oxid Med Cell Longev. 2017;2017:1703810. doi:10.1155/2017/1703810
[3] Foresti R, Bains SK, Pitchumony TS, et al. Small molecule activators of the Nrf2-HO-1 antioxidant axis modulate heme metabolism and inflammation in BV2 microglia cells. Pharmacol Res. 2013;76:132-148. doi:10.1016/j.phrs.2013.07.010
[4] Ali I, Li C, Kuang M, et al. Nrf2 Activation and NF-Kb & caspase/bax signaling inhibition by sodium butyrate alleviates LPS-induced cell injury in bovine mammary epithelial cells. Mol Immunol. 2022;148:54-67. doi:10.1016/j.molimm.2022.05.121
- Changes in discussion part, P. 18, ll: 617-620: ” Furthermore, the analysis should focus on the interaction of other signaling pathways, such as nuclear factor- (erythroid-derived 2-) like 2 (Nrf2, antioxidant response [92,93]) in mediating the control of apoptosis and endothelial cell recovery to understand the complex interplay.”.
- We added 2 new references:
- Wang, M.X.; Zhao, J.; Zhang, H.; Li, K.; Niu, L.Z.; Wang, Y.P.; Zheng, Y.J. Potential protective and therapeutic roles of the nrf2 pathway in ocular diseases: an update. Oxid Med Cell Longev. 2020, 2020, 9410952. https://doi.org/10.1155/2020/9410952.
- Batliwala, S.; Xavier, C.; Liu, Y.; Wu, H.; Pang, I.H. Involvement of Nrf2 in ocular diseases. Oxid Med Cell Longev. 2017, 2017, 1703810. https://doi.org/10.1155/2017/1703810.
Reviewer1: Minor Concerns:
Line 26: change preventing CE to inhibiting
Line 443: change will to was used
- Authors: We would like to thank the reviewer for these helpful hints. We changed accordingly:
- 1, ll 25-26: “In this study, we demonstrated that suppressing pro-apoptotic siRNA leads to inhibiting CE apoptosis”
- 14, ll 446-448: “In order to test the hypothesis that pro-apoptotic gene transfer can promote corneal endothelial survival in an eye bank environment, the following experiment was used to analyze the apoptosis rate in donor corneas “
Reviewer 2 Report
Comments and Suggestions for Authors
The article "Suppressing pro-apoptotic proteins by siRNA in corneal endo-2 thelial cells protects against cell death" from Staehlke et al, investigates the potential gene therapy with anti-apoptotic siRNAs in corneas in storage with the aim to prolong survival. The study is of great importance due to the shortage of high quality corneas suitable for transplantation worldwide. The use of siRNA will offer silent protection in contrast of virus expressing shRNA, however I understand that this is an initial study.
The paper is well written and results are clearly presented.
I recommend Accept after minor revision (correction of minor methodological errors and text editing)
Corrections:
Figure S1 legend. Line 5. "...pro-apototoc protein Bax or Bax". It should be "Bax or Bak".
Figure 4 legend. I suggest changing "In contrast, Bax+Bak-siRNA transfected HCECs show the same morphology as cells without apoptosis stimulus (W/o inducers)" for "In contrast, Bax+Bak-siRNA transfected HCECs and treated with apoptosis inducers have improved morphology, resembling cells without apoptosis stimulus (W/o inducers)"
I suggest changing "In contrast, a clear continuous ZO-1 band was found in the Bax+Bak-siRNA transfected cells and in the cells without apoptosis induction, indicating an intact vital cell layer" for "In contrast, a clear continuous ZO-1 band was found in the Bax+Bak-siRNA transfected cells and treated with etoposide. The cell junction protection of Bax+Bak-siRNA was less evident for staurosporin"
Figure 6 Legend I suggest to change "...and tight cell-cell contact of HCECs can be observed, comparable to endothelial cells without (W/o) apoptosis induction" for "...and tight cell-cell contact of HCECs can be observed, comparable to endothelial cells without (W/o) apoptosis induction for etoposide and to a less extent for staurosporin".
Author Response
Reviewer 2:
Biomedicines-3058990
Suppressing pro-apoptotic proteins by siRNA in corneal endothelial cells protects against cell death
Staehlke et al.
Comments and Suggestions for Authors
The article "Suppressing pro-apoptotic proteins by siRNA in corneal endothelial cells protects against cell death" from Staehlke et al, investigates the potential gene therapy with anti-apoptotic siRNAs in corneas in storage with the aim to prolong survival. The study is of great importance due to the shortage of high quality corneas suitable for transplantation worldwide. The use of siRNA will offer silent protection in contrast of virus expressing shRNA, however I understand that this is an initial study.
The paper is well written and results are clearly presented.
I recommend Accept after minor revision (correction of minor methodological errors and text editing)
Reviewer2: Corrections:
Authors: We would like to thank you for these helpful tips and comments. With their formulation, the documentation is now more accurate and relevant.
- Reviewer2: Figure S1 legend. Line 5. "...pro-apototoc protein Bax or Bax". It should be "Bax or Bak".
- Author: Thanks for the hint. Supplementary file: Line 5 “ pro-apoptotic Bax or Bak.”
- Reviewer2: Figure 4 legend. I suggest changing "In contrast, Bax+Bak-siRNA transfected HCECs show the same morphology as cells without apoptosis stimulus (W/o inducers)" for "In contrast, Bax+Bak-siRNA transfected HCECs and treated with apoptosis inducers have improved morphology, resembling cells without apoptosis stimulus (W/o inducers)"
- Authors: The reviewer is right to point out a misleading formulation of this aspect. We have revised this sentence. Figure 4: p. 8, ll:341-342: “In contrast, Bax+Bak-siRNA transfected HCECs and treated with apoptosis inducers have improved morphology, resembling cells without apoptosis stimulus (W/o inducers).”
- Reviewer2: I suggest changing "In contrast, a clear continuous ZO-1 band was found in the Bax+Bak-siRNA transfected cells and in the cells without apoptosis induction, indicating an intact vital cell layer" for "In contrast, a clear continuous ZO-1 band was found in the Bax+Bak-siRNA transfected cells and treated with etoposide. The cell junction protection of Bax+Bak-siRNA was less evident for staurosporin"
- Authors: Thanks for the advice. We have changed in results p. 10 ll:384-386: “In contrast, a clear continuous ZO-1 band was found in the Bax+Bak-siRNA transfected cells after treatment with etoposide, and in the cells without apoptosis induction, indicating an intact vital cell layer. The cell junction protection of Bax+Bak-siRNA was less evident for staurosporin.”
- Reviewer2: Figure 6 Legend I suggest to change "...and tight cell-cell contact of HCECs can be observed, comparable to endothelial cells without (W/o) apoptosis induction" for "...and tight cell-cell contact of HCECs can be observed, comparable to endothelial cells without (W/o) apoptosis induction for etoposide and to a less extent for staurosporin".
Authors: Thanks for the advice. We have changed in legend of Figure 6: p. 11, ll: 405-408: “. In contrast, almost no TUNEL-positive cells are found in Bax+Bak-siRNA-transfected cells, and tight cell-cell contact of HCECs can be observed, comparable to endothelial cells without (W/o) apoptosis induction for etoposide and to a less extent for staurosporine.”
Round 2
Reviewer 1 Report
Comments and Suggestions for Authors
The authors have performed an exemplary study that is flawless. I have no other suggestions to offer.